# Set-Valued Prediction in Hierarchical Classification with Constrained Representation Complexity

**Thomas Mortier**[1]       **Eyke Hüllermeier**[2]       **Krzysztof Dembczyński**[3,4]       **Willem Waegeman**[1]

[1]Dept. of Data Analysis and Mathematical Modelling, Ghent University, Coupure links 653, Ghent, Belgium
[2]Institute of Informatics, LMU Munich, Akademiestr. 7, Munich, Germany
[3]Institute of Computing Science, Poznań University of Technology, Piotrowo 2, Poznań, Poland
[4]Yahoo! Research, 770 Broadway, New York, USA

## Abstract

Set-valued prediction is a well-known concept in multi-class classification. When a classifier is uncertain about the class label for a test instance, it can predict a set of classes instead of a single class. In this paper, we focus on hierarchical multi-class classification problems, where valid sets (typically) correspond to internal nodes of the hierarchy. We argue that this is a very strong restriction, and we propose a relaxation by introducing the notion of representation complexity for a predicted set. In combination with probabilistic classifiers, this leads to a challenging inference problem for which specific combinatorial optimization algorithms are needed. We propose three methods and evaluate them on benchmark datasets: a naïve approach that is based on matrix-vector multiplication, a reformulation as a knapsack problem with conflict graph, and a recursive tree search method. Experimental results demonstrate that the last method is computationally more efficient than the other two approaches, due to a hierarchical factorization of the conditional class distribution.

## 1 INTRODUCTION

In multi-class classification problems with a lot of classes, there are often situations where a classifier is uncertain about the class label for a given instance, e.g., because of class ambiguity. Set-valued predictions form a natural way of dealing with this uncertainty, by predicting a set of classes instead of a single class. For instance, in medical diagnosis, when there is uncertainty related to the true disease of a patient, a set-valued classifier will return a set of candidate diseases. This set can then be of great help for a medical doctor, as only the remaining candidate diseases need further investigation.

In the machine learning literature, set-valued prediction has been studied under different frameworks. A simple approach consists of top-$k$ prediction, i.e., returning a set with the $k$ classes that have the highest probabilities or scores [Lapin et al., 2016, Chzhen et al., 2021]. Another popular approach is conformal prediction [Shafer and Vovk, 2008], which produces sets that contain the true class with high probability. A third framework is rooted in Bayesian decision theory and optimizes a utility function that trades off two important criteria for set-valued predictions, namely correctness and precision [Del Coz et al., 2009, Corani and Zaffalon, 2008, 2009, Zaffalon et al., 2012, Yang et al., 2017b, Mortier et al., 2021]. Like in conformal prediction, the predicted set should be correct in the sense of covering the true class, but at the same time, the set should be precise and not contain too many options.

Set-valued prediction has also been considered in a hierarchical classification setting, where similarity among classes is encoded by means of a predefined class hierarchy provided by domain experts. For instance, in medical diagnosis, it is natural to group different types of cancer as one branch of the disease classification hierarchy. In hierarchical classification, set-valued predictions are often restricted to specific subsets of the set of classes, namely those that correspond to nodes of the hierarchy and, therefore, have a clear interpretation and are deemed semantically meaningful [Alex Freitas, 2007, Bi and Kwok, 2015, Rangwala and Naik, 2017, Yang et al., 2017a]. Moreover, restricting candidate sets to hierarchy nodes will also reduce the computational complexity of finding the best prediction for a given instance. On the other side, a restriction of that kind may negatively impact predictive performance. That's why a few authors allow any subset of classes as a prediction in hierarchical classification [Oh, 2017, Mortier et al., 2021]. Then, however, predictions might be semantically questionable and, moreover, difficult to communicate – in the general case, a prediction would be an enumeration of (possibly many) leaf nodes, ignoring the hierarchy altogether.

In this paper, we propose a novel set-valued prediction

*Accepted for the 38th Conference on Uncertainty in Artificial Intelligence* (UAI 2022).

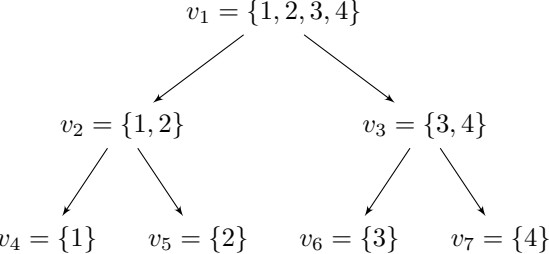

$$v_1 = \{1, 2, 3, 4\}$$

$$v_2 = \{1, 2\} \qquad v_3 = \{3, 4\}$$

$$v_4 = \{1\} \qquad v_5 = \{2\} \qquad v_6 = \{3\} \qquad v_7 = \{4\}$$

Figure 1: Example hierarchy for $\mathcal{Y} = \{1, 2, 3, 4\}$. The class space is represented by the root of the tree structure $\mathcal{T}$, given by $v_1$. For $\hat{Y} = \{3, 4\}$ we find $\mathcal{S}_{\mathcal{T}}(\hat{Y}) = \{\{v_6, v_7\}, \{v_3\}\}$ and therefore $R_{\mathcal{T}}(\hat{Y}) = 1$.

framework for hierarchical classification that makes a compromise between the two extremes. Compared to approaches that predict a single node of the hierarchy, we will be less restrictive in the type of sets that can be returned, but we will be more restrictive than methods that return any subset of classes. More specifically, we allow the user to restrict the so-called *representation complexity* of a predicted set (see Section 2 for a formal definition). The main idea is to return a restricted number of internal nodes of the hierarchy as candidate sets instead of a single node. For example, imagine that classes correspond to spatial regions on the earth. In this case, a natural hierarchy is the form of

$$\text{continent} \rightarrow \text{country} \rightarrow \text{state} \rightarrow \text{district} \rightarrow \cdots.$$

Obviously, one is interested in a prediction that makes it probable to find the right location. In "flat" top-$k$ prediction, one may end up with many small regions (leaf nodes of the hierarchy) scattered around the globe, which might not be desirable (e.g., checking those regions may cause a lot of effort). On the contrary, hierarchical predictions such as "it's in France or in the Netherlands" might be more useful and require less effort.

Section 2 presents a decision-theoretic framework where the representation complexity of a set is a user-defined parameter, which results in a challenging optimization problem. In Section 3, we present three different approaches to solve this inference problem: a naïve algorithm that has a high computational complexity, a reformulation as a knapsack problem with conflict graph, and a tailored recursive tree search algorithm that adopts a hierarchical factorization of the conditional class distribution. In Section 4, we discuss related work, and in Section 5, we present experimental results on five challenging hierarchical classification datasets.

## 2 FORMAL PROBLEM FORMULATION

In a standard multi-class classification setting we assume that training and test data are i.i.d. according to an unknown

distribution $P(\boldsymbol{x}, y)$ on $\mathcal{X} \times \mathcal{Y}$, with $\mathcal{X}$ some instance space (e.g., images, documents, etc.) and $\mathcal{Y} = \{c_1, \ldots, c_K\}$ a class space consisting of $K$ classes. In a multi-class classification setting, we estimate the conditional class probabilities $P(\cdot \mid \boldsymbol{x})$ over $\mathcal{Y}$, with properties $\forall c \in \mathcal{Y} : 0 \leq P(c \mid \boldsymbol{x}) \leq 1, \sum_{c \in \mathcal{Y}} P(c \mid \boldsymbol{x}) = 1$. This distribution can be estimated using a wide range of well-known probabilistic methods, such as logistic regression, linear discriminant analysis, gradient boosting trees or neural networks with a softmax output layer. At prediction time, we will predict sets $\hat{Y}$ that are subsets of $\mathcal{Y}$. The probability mass of such a set will be computed as $P(\hat{Y} \mid \boldsymbol{x}) = \sum_{c \in \hat{Y}} P(c \mid \boldsymbol{x})$.

However, in this paper we will consider a hierarchical multi-class classification setting. Hence, we assume that a domain expert has defined a hierarchy over the class space, in the form of a tree structure $\mathcal{T}$ that contains in general $M$ nodes. $\mathcal{V}_{\mathcal{T}} = \{v_1, \ldots, v_M\}$ will denote the set of nodes and every node identifies a set of classes. As special cases, the root $v_1$ represents the class space $\mathcal{Y}$, and the leaves represent individual classes – see Fig. 1 for a simple example. In hierarchical classification, one typically makes the strong restriction $\hat{Y} \in \mathcal{V}_{\mathcal{T}}$ for predicted sets – see e.g., Bi and Kwok [2015]. The probability mass $P(v \mid \boldsymbol{x})$ of such a set can be computed using the chain rule of probability:

$$P(v \mid \boldsymbol{x}) = \prod_{v' \in \text{Path}(v)} P(v' \mid \text{Parent}(v'), \boldsymbol{x}), \quad (1)$$

where $\text{Path}(v)$ is a set of nodes on the path connecting the node $v$ and the root of the tree structure. $\text{Parent}(v)$ gives the parent of node $v$, and for the root node $v_1$ we have $P(v_1 \mid \text{Parent}(v_1), \boldsymbol{x}) = 1$. In each node of the tree, one can train any multi-class probabilistic classifier. Classical models of that kind include nested dichotomies [Fox, 1997, Frank and Kramer, 2004, Melnikov and Hüllermeier, 2018], conditional probability estimation trees [Beygelzimer et al., 2009] and probabilistic classifier trees [Dembczyński et al., 2016]. In neural networks with a hierarchical softmax output layer, all nodes are trained simultaneously [Morin and Bengio, 2005].

In this work, we do not focus on the training algorithms. Instead we assume that a probabilistic model has been estimated, either with classical models or using a hierarchical factorization as in Eqn. (1), and we present a decision-theoretic framework with an inference procedure at prediction time. In this inference procedure, we restrict the representation complexity $R_{\mathcal{T}}(\hat{Y})$, which will be formally defined as the minimal number of tree nodes needed to represent the set $\hat{Y}$. Let $\mathcal{S}_{\mathcal{T}}(\hat{Y})$ denote the set of all disjoint combinations of tree nodes that represent $\hat{Y}$:

$$\mathcal{S}_{\mathcal{T}}(\hat{Y}) = \left\{ \hat{V} \subset \mathcal{V}_{\mathcal{T}} : \bigcup_{v_i \in \hat{V}} v_i = \hat{Y} \wedge \bigcap_{v_i \in \hat{V}} v_i = \emptyset \right\}.$$

Then, we define the representation complexity of the predic-

tion $\hat{Y}$ as

$$R_{\mathcal{T}}(\hat{Y}) = \min_{\hat{V} \in \mathcal{S}_{\mathcal{T}}(\hat{Y})} |\hat{V}|, \qquad (2)$$

with $|\hat{V}|$ the cardinality of $\hat{V}$. As an example, let us consider again the four-class hierarchy that was shown in Fig. 1. For example, with $\hat{Y} = \{c_1, c_3, c_4\}$ we find $\mathcal{S}_{\mathcal{T}}(\hat{Y}) = \{\{v_4, v_6, v_7\}, \{v_4, v_3\}\}$ and therefore $R_{\mathcal{T}}(\hat{Y}) = 2$.

Furthermore, if we denote the $r$-th representation complexity class by

$$\mathcal{R}_{\mathcal{T}}^{(r)} = \left\{ \hat{Y} \in \mathcal{P}(\mathcal{Y}) : R_{\mathcal{T}}(\hat{Y}) = r \right\},$$

then it immediately folllows that $\mathcal{R}_{\mathcal{T}}^{(1)} = \mathcal{V}_{\mathcal{T}}$. In the example of Fig. 1, the other representation complexity classes are given by:

$$\mathcal{R}_{\mathcal{T}}^{(2)} = \{\{1, 3\}, \{1, 4\}, \{2, 3\}, \{2, 4\}, \{1, 3, 4\}, \{2, 3, 4\},$$
$$\{1, 2, 3\}, \{1, 2, 4\}\}, \qquad \mathcal{R}_{\mathcal{T}}^{(3)} = \{\emptyset\}.$$

The example suggests that the first $K - 1$ representation complexity classes form a partition of $\mathcal{P}(\mathcal{Y}) \setminus \{\emptyset\}$, with $\mathcal{P}(\mathcal{Y})$ the powerset of $\mathcal{Y}$. The following theorem, whose proof is found in App. A, indicates that this observation holds more generally.

**Theorem 1.** $\{\mathcal{R}_{\mathcal{T}}^{(1)}, \ldots, \mathcal{R}_{\mathcal{T}}^{(K-1)}\}$ *forms a partition of* $\mathcal{P}(\mathcal{Y}) \setminus \{\emptyset\}$, *for any class space* $\mathcal{Y}$ *and hierarchy* $\mathcal{T}$.

We are now ready to introduce the inference problem that forms the central idea of this paper. At prediction time, we aim to find the set with highest probability mass, while restricting the maximal representation complexity by $r$ and the maximal set size by $k$, with $r$ and $k$ user-defined parameters. As a result, we aim to solve the following constrained maximization problem:

$$\hat{Y}^*(\boldsymbol{x}) = \arg\max_{\hat{Y} \subseteq \mathcal{Y}} P(\hat{Y} \mid \boldsymbol{x}), \qquad (3)$$

$$\text{subject to} \qquad |\hat{Y}| \leq k, \quad R_{\mathcal{T}}(\hat{Y}) \leq r,$$

where $|\hat{Y}|$ denotes the cardinality of the predicted set $\hat{Y}$. Remark that in classical hierarchical classification settings, one would have the very tight restriction $R_{\mathcal{T}}(\hat{Y}) = 1$, whereas in flat classification $R_{\mathcal{T}}(\hat{Y}) \leq K$ typically applies.

## 3 ALGORITHMS

In this section we will discuss three algorithms that can be used to solve problem (3), which is a very challenging combinatorial optimization problem, because the number of feasible sets grows exponentially with $r$. To this end, we will assume that we have access to an estimate of the conditional class distribution $P(\cdot \mid \boldsymbol{x})$. For the first two algorithms that

we present, such an estimate can be obtained using any probabilistic classifier. For the third algorithm, a specific hierarchical factorization as in Eqn. (1) is needed. Owing to this factorization, we obtain substantial improvements in memory and runtime complexity.

### 3.1 MATRIX-VECTOR MULTIPLICATION

A naïve (but inefficient) algorithm performs an exhaustive search over all feasible solutions of problem (3). By relying on fast matrix-vector multiplication and parallelization routines, this can still be done within reasonable time for small $r$. Assuming that $r$ and $k$ remain fixed, let us denote the set of feasible solutions by

$$\mathcal{M}_{r,k} = \left\{ \hat{Y} \in \mathcal{P}(\mathcal{Y}) : R_{\mathcal{T}}(\hat{Y}) \leq r \wedge |\hat{Y}| \leq k \right\}.$$

Given this set, together with some arbitrary ordering, let us further consider a matrix $\boldsymbol{M} \in \{0, 1\}^{|\mathcal{M}| \times K}$ where rows represent the elements of $\mathcal{M}_{r,k}$ and columns the elements $\mathcal{Y}$. In this matrix, element $M_{i,j} = 1$ if the $i$-th set in $\mathcal{M}_{r,k}$ contains class $c_j$. For a given $\boldsymbol{x}$, let us denote by $\boldsymbol{p}$ the vector containing conditional class probabilities, i.e., $p_j = P(c_j \mid \boldsymbol{x})$. The solution to (3) is then simply found by calculating the vector $\boldsymbol{Mp}$ and searching for the highest element in this vector, as shown in Alg. 1. As a consequence of Theorem 1, it is clear that the runtime and memory complexity for this naïve algorithm rapidly increases as a function of $r$. The complexity is of the order $O(2^K)$ in the worst case, when $r$ is close to $K$.

---

**Algorithm 1** MVM – **input:** $\boldsymbol{x}, \mathcal{M}_{r,k}, \boldsymbol{M}, P, \mathcal{Y}$

---

1: $\hat{Y}^*, p_{\hat{Y}^*} \leftarrow \emptyset, 0$
2: $\boldsymbol{p} \leftarrow$ conditional class probabilities, i.e., $p_j = P(c_j \mid \boldsymbol{x})$
3: $\boldsymbol{p}_{\mathcal{M}} \leftarrow \boldsymbol{Mp}$ with $p_{\mathcal{M},\hat{Y}} = P(\hat{Y} \mid \boldsymbol{x})$ for $\hat{Y} \in \mathcal{M}_{r,k}$
4: **for** $\hat{Y} \in \mathcal{M}_{r,k}$ **do**
5:     **if** $p_{\mathcal{M},\hat{Y}} \geq p_{\hat{Y}^*}$ **then**
6:         $\hat{Y}^*, p_{\hat{Y}^*} \leftarrow \hat{Y}, p_{\mathcal{M},\hat{Y}}$
7: **return** $\hat{Y}^*, p_{\hat{Y}^*}$

---

### 3.2 KNAPSACK WITH CONFLICT GRAPH

A second algorithm consists of reducing (3) to an instance of the knapsack problem with conflict graph (KCG) [Pferschy and Schauer, 2009]. In our case, items are represented by tree nodes, where every tree node is either included in the knapsack or not. The goal is then to find the set of nodes that maximize the total probability mass, while taking into account the constraints on the representation complexity and the set size. In addition, we also have constraints w.r.t. incompatibility of certain pairs of nodes. More precisely, for any pair of tree nodes, where one node of the pair is an

ancestor of the other node, at most one node can be included in the knapsack. This can be represented by means of an undirected conflict graph, where every vertex corresponds to a tree node and every edge denotes a conflict relation. More formally, to translate our problem to an instance of KCG, let us first denote by $\mathcal{G}_{\mathcal{T}} = (\mathcal{V}_{\mathcal{T}}, \mathcal{E}_{\mathcal{T}})$ the conflict graph with

$$\mathcal{E}_{\mathcal{T}} = \{(v_i, v_j) : (v_i, v_j) \in \mathcal{V}_{\mathcal{T}} \times \mathcal{V}_{\mathcal{T}} \wedge v_i \cap v_j \neq \emptyset\} \,.$$

For every edge $(v_i, v_j) \in \mathcal{E}_{\mathcal{T}}$, we have a corresponding vector representation given by $\boldsymbol{e} \in \{0, 1\}^{|\mathcal{V}_{\mathcal{T}}|}$ with $e_i = e_j = 1$ and $\sum_{j=1}^{|\mathcal{V}_{\mathcal{T}}|} e_j = 2$. Furthermore, let us denote by $\boldsymbol{w}$ the $|\mathcal{V}_{\mathcal{T}}|$-dimensional vector that encodes for every tree node the size of the corresponding set of classes, i.e., $w_j = |v_j|$. For a given $\boldsymbol{x}$, let $\boldsymbol{p}$ be the $|\mathcal{V}_{\mathcal{T}}|$-dimensional vector that contains the probability mass $P(v_j \mid \boldsymbol{x})$ of every tree node $v_j$. Let $\boldsymbol{z} \in \{0, 1\}^{|\mathcal{V}_{\mathcal{T}}|}$ be the vector that encodes feasible solutions, i.e., an entry in this vector is 1 when the corresponding node is contained in the knapsack, and 0 otherwise. Given the above notations, the solution to (3) is then found by solving the following integer linear program (ILP):

$$\max_{\boldsymbol{z}} \quad \boldsymbol{p}^{\mathsf{T}} \boldsymbol{z}, \quad \text{subject to} \quad \boldsymbol{A}_{\mathcal{T}} \boldsymbol{z} \leq \boldsymbol{b}_{r,k} \,, \tag{4}$$

with

$$\boldsymbol{A}_{\mathcal{T}} = \begin{bmatrix} \boldsymbol{1} & \boldsymbol{w} & \boldsymbol{e}_1 & \ldots & \boldsymbol{e}_{|\mathcal{E}_{\mathcal{T}}|} \end{bmatrix}^{\mathsf{T}},$$
$$\boldsymbol{b}_{r,k} = \begin{bmatrix} r & k & 1 & \ldots & 1 \end{bmatrix}^{\mathsf{T}}.$$

Alg. 2 describes the full procedure to obtain the Bayes-optimal solution, using a generic ILP solver. It will be faster than Alg. 1, but long runtimes can still be expected, because KCG problems are known as strongly NP-hard problems [Pferschy and Schauer, 2009]. In the related work section, we give an overview of algorithms that have been developed for this group of problems.

---

**Algorithm 2** KCG – input: $\boldsymbol{x}$, $\boldsymbol{A}_{\mathcal{T}}$, $\boldsymbol{b}_{r,k}$, ILP, $P$, $\mathcal{V}_{\mathcal{T}}$

---

1: $\hat{Y}^*, p_{\hat{Y}^*} \leftarrow \emptyset, 0$
2: Compute $\boldsymbol{p}$ using an estimated probabilistic model $P(\cdot \mid \boldsymbol{x})$
3: $\hat{Y}^*, p_{\hat{Y}^*} \leftarrow \text{ILP}(\boldsymbol{p}, \boldsymbol{A}_{\mathcal{T}}, \boldsymbol{b}_{r,k}) \triangleright$ Solve with a given ILP solver
4: **return** $\hat{Y}^*, p_{\hat{Y}^*}$

---

### 3.3 RECURSIVE TREE SEARCH

The last algorithm is a tailor-made recursive tree search, inspired by $A^*$-search for probabilistic classifier trees [Dembczyński et al., 2012, 2016, Mena et al., 2017, Mortier et al., 2021]. Unlike the previous two approaches, which are both usable with flat and hierarchical classifiers, this method assumes that the conditional class distribution can be factorized as in Eqn. (1). This restriction will result in significant

speed-ups, because the search for the Bayes-optimal solution of problem (3) can then be performed in a top-down manner.

At its core, the algorithm uses a priority queue for storing visited nodes in decreasing order of probability mass. First, the queue is initialized with the root node in Alg. 3. Next, in the main loop of Alg. 4, for each iteration, the next node is popped from the priority queue in order of decreasing probability mass. For each node that is popped, the current solution is updated and compared to the best solution seen so far. Subsequently, Alg. 4 is recursively called with a copy of the priority queue. In this way, solutions are recursively explored in a depth-first search manner until the maximum level (i.e., representation complexity $r$) is reached. To show that Alg. 4 finds the Bayes-optimal solution to problem (3) in an efficient way, we first prove that the equality $\hat{Y} \cap v = \emptyset$ must hold for any feasible set $\hat{Y} \cup v$ considered in line 3 (Prop. 1). Subsequently, we show in Theorem 2 that from those feasible sets, only a limited number needs to be considered to find the Bayes-optimal solution.

**Proposition 1.** *For any solution $\hat{Y} \in \mathcal{P}(\mathcal{Y})$ and corresponding priority queue $\mathcal{Q}_{\hat{Y}}$ in Alg. 4, there are no nodes $v$ in $\mathcal{Q}_{\hat{Y}}$ for which $\hat{Y} \cap v \neq \emptyset$ in line 3. This holds for any $\boldsymbol{x}, r, k, P$ and $\mathcal{V}_{\mathcal{T}}$.*

*Proof.* The proposition holds naturally for the first call of Alg. 4, i.e., when $\hat{Y} = \emptyset$. Let us now consider all $\hat{Y}$ for which $R_{\mathcal{T}}(\hat{Y}) = 1$, or in other words all $\hat{Y} \in \mathcal{V}_{\mathcal{T}}$. Furthermore, assume that there exists a $v \in \mathcal{Q}_{\hat{Y}}$ such that $\hat{Y} \cap v \neq \emptyset$, then $v$ must be one of the descendants of $\hat{Y}$ or vice versa. The first case is not possible, since descendants of $\hat{Y}$ are only added to $\mathcal{Q}_{\hat{Y}}$ after the recursive call in line 10 has finished. Nor is the second, since $v$ must be already popped from the priority queue in that case. Therefore, the above proposition must hold for any $\hat{Y}$ with $R_{\mathcal{T}}(\hat{Y}) = 1$. Let us now assume that the proposition holds for any $\hat{Y}$ for which $R_{\mathcal{T}}(\hat{Y}) = r' < r$. Assume that for all $\hat{Y}$ with $R_{\mathcal{T}}(\hat{Y}) = r' + 1$ there exists a $v$ in $\mathcal{Q}_{\hat{Y}}$ such that $\hat{Y} \cap v \neq \emptyset$. Since there exists a $v''$ in $\mathcal{V}_{\mathcal{T}}$ such that $\hat{Y} = \hat{Y}'' \cup v''$ with $\mathcal{Q}_{\hat{Y}''} = \mathcal{Q}_{\hat{Y}} \cup v''$, then either $v'' \cap v \neq \emptyset$ or $\hat{Y}'' \cap v \neq \emptyset$. Similarly as in the beginning of the proof, the first case is not possible since descendants of $v''$ are only added to the priority queue after the recursive call in line 10. For the second case, given that the proposition holds for any $\hat{Y}$ with $R_{\mathcal{T}}(\hat{Y}) = r' < r$, we know that there is no $v \in \mathcal{Q}_{\hat{Y}}$ with $\hat{Y}'' \cap v \neq \emptyset$. This contradiction completes the proof by induction. $\square$

**Theorem 2.** *For any $\boldsymbol{x}, r, k, P$ and $\mathcal{V}_{\mathcal{T}}$, Alg. 3 will find the Bayes-optimal solution of problem (3).*

*Proof.* Without lines 4–6,8,11–13 and 17–18, it is clear that Alg. 4 will visit all sets in $\mathcal{R}_{\mathcal{T}}^{(1)} \cup \ldots \cup \mathcal{R}_{\mathcal{T}}^{(r)}$. The check in line 4 makes sure that only sets that satisfy the size

constraint are 1) compared with the best solution so far and 2) are considered as current solution for a next recursive call in line 10. Moreover, with respect to the latter, if for the current solution we have that $|\hat{Y}| = k$, then we are not allowed to include additional nodes, hence, the additional check in line 8. Furthermore, we can return to the parent call in line 12 since the maximum level (i.e., representation complexity $r$) is reached and for any subsequent node $v'$ that is popped from $\mathcal{Q}_{\hat{Y}}$ we know that $P(\hat{Y} \cup v' \mid \boldsymbol{x}) \leq P(\hat{Y}' \mid \boldsymbol{x})$. Finally, for any level of the recursion, we can also return to the parent call in line 18 from the moment that we pop a leaf node. Indeed, assume that for a given level of recursion and iteration in the while loop of Alg. 4, the first leaf node $v_l$ is popped from $\mathcal{Q}_{\hat{Y}}$, resulting in a new candidate solution $\hat{Y}_l = \hat{Y} \cup v_l$. Let us denote by $v_n$ and $\hat{Y}_n = \hat{Y} \cup v_n$ any subsequent node that is popped from $\mathcal{Q}_{\hat{Y}}$ and corresponding candidate solution. We have to show that there is no solution containing $\hat{Y}_n$ having a strictly higher probability mass than all solutions containing $\hat{Y}_l$. Let's assume that there exists a solution that satisfies the above, which we denote by $\hat{Y}'_n = \hat{Y}_n \cup \hat{V}$ with $\hat{V} \subset \mathcal{V}_{\mathcal{T}}$. With a similar reasoning as in Prop. 1, we know that $v_l \cap \hat{V} = \emptyset$, and hence, the solution $\hat{Y}'_l = \hat{Y}_l \cup \hat{V}$ must also be visited by Alg. 4. Taking into account the property of the priority queue we know that:

$$P(\hat{Y}_l \mid \boldsymbol{x}) \geq P(\hat{Y}_n \mid \boldsymbol{x}) \Leftrightarrow P(\hat{Y}'_l \mid \boldsymbol{x}) \geq P(\hat{Y}'_n \mid \boldsymbol{x}),$$

which is in contrast with the above, and therefore, completes the proof by contradiction. □

---

**Algorithm 3** RTS – **input:** $\boldsymbol{x}, r, k, P, \mathcal{V}_{\mathcal{T}}$

---
1: $\mathcal{Q} = \emptyset$
2: $\mathcal{Q}.\text{add}((v_1, 1))$
3: $\hat{Y}^*, p_{\hat{Y}*} \leftarrow \text{RTS.find}(\boldsymbol{x}, r, k, \emptyset, 0, \emptyset, 0, \mathcal{Q}, P, \mathcal{V}_{\mathcal{T}})$
4: **return** $\hat{Y}^*, p_{\hat{Y}*}$

---

Taking into account the stopping criterion in line 18 of Alg. 4 that is proven in Theorem 2, while assuming a complete binary tree with depth $\log_2 K$ as hierarchy $\mathcal{T}$, an upper bound on the time complexity of Alg. 3 is therefore given by $O(\log_2 K^r)$.

# 4  RELATED WORK

In flat multi-class classification, similar inference problems as problem (3) are considered, but without any restrictions on the representation complexity. This setting is simply referred to as top-$k$ prediction, and very popular in applied papers, e.g., papers that report the recall@$k$. A few other authors who study top-$k$ prediction in a more fundamental way prove that the top-$k$ can simply be found by the $k$ classes with the highest conditional class probabilities [Lapin et al.,

---

**Algorithm 4** RTS.find – **input:** $\boldsymbol{x}, r', k, \hat{Y}^*, p_{\hat{Y}*}, \hat{Y}, p_{\hat{Y}}, \mathcal{Q}_{\hat{Y}}, P, \mathcal{V}_{\mathcal{T}}$

---
1: **while** $\mathcal{Q}_{\hat{Y}} \neq \emptyset$ **do**
2: $\quad (v, p_v) \leftarrow \mathcal{Q}_{\hat{Y}}.\text{pop}()$
3: $\quad \hat{Y}', p_{\hat{Y}'} \leftarrow \hat{Y} \cup v, p_{\hat{Y}} + p_v$
4: $\quad$ **if** $|\hat{Y}'| \leq k$ **then**
5: $\quad\quad$ **if** $p_{\hat{Y}'} \geq p_{\hat{Y}*}$ **then**
6: $\quad\quad\quad \hat{Y}^*, p_{\hat{Y}*} \leftarrow \hat{Y}', p_{\hat{Y}'}$
7: $\quad\quad$ **if** $r' > 1$ **then**
8: $\quad\quad\quad$ **if** $|\hat{Y}'| \neq k$ **then**
9: $\quad\quad\quad\quad \mathcal{Q}_{\hat{Y}'} \leftarrow \mathcal{Q}_{\hat{Y}}$ $\quad$ ▷ Copy priority queue
10: $\quad\quad\quad\quad \hat{Y}^*, p_{\hat{Y}*} \leftarrow \text{RTS.find}(\boldsymbol{x}, r' - 1, k, \hat{Y}^*,$ $p_{\hat{Y}*}, \hat{Y}', p_{\hat{Y}'}, \mathcal{Q}_{\hat{Y}'}, P, \mathcal{V}_{\mathcal{T}})$
11: $\quad\quad\quad$ **else**
12: $\quad\quad\quad\quad$ **break**
13: $\quad\quad$ **if** $v$ is not a leaf node **then**
14: $\quad\quad\quad$ **for** $v' \in \text{Children}(v)$ **do**
15: $\quad\quad\quad\quad p_{v'} \leftarrow p_v \times P(v' \mid v, \boldsymbol{x})$
16: $\quad\quad\quad\quad \mathcal{Q}_{\hat{Y}}.\text{add}((v', P(v' \mid \boldsymbol{x})))$
17: $\quad\quad$ **else**
18: $\quad\quad\quad$ **break**
19: **return** $\hat{Y}^*, p_{\hat{Y}*}$

---

2016, Chzhen et al., 2021]. Top-$k$ predictions are also frequently used in the context of extreme multi-label classification, where the number of labels is very large [Prabhu and Varma, 2014, Babbar and Schölkopf, 2017, Prabhu et al., 2018, Wydmuch et al., 2018, Zhuo et al., 2020, Chang et al., 2020].

Authors such as Chzhen et al. [2021] refer to top-$k$ prediction as pointwise size control. They also discuss many other set-valued prediction settings, including average size control [Denis and Hebiri, 2017], average error control (such as conformal prediction) [Sadinle et al., 2019, Lei, 2014, Shafer and Vovk, 2008] and pointwise error control [Cai et al., Lei and Wasserman, 2014, Vovk, 2012]. Another set-valued prediction framework for flat multi-class classification is rooted in Bayesian decision theory and optimizes a utility function that trades off the two important criteria for set-valued predictions, namely correctness and precision [Del Coz et al., 2009, Corani and Zaffalon, 2008, 2009, Zaffalon et al., 2012, Yang et al., 2017b, Mortier et al., 2021].

Set-valued prediction has also been considered in hierarchical multi-class classification. Here, too, various frameworks exist, which typically differ in the type of loss function that is considered, and in the flexibility in representation complexity that is allowed. Many papers restrict the representation complexity of the predicted set to one, using abstention strategies for classifiers in internal nodes of the hierarchy [Alex Freitas, 2007, Rangwala and Naik, 2017, Yang et al.,

2017a]. For example, Sun and Lim [2001] propose a simple stopping strategy based on thresholding. When the probability mass for a given node is greater than a predefined threshold, the sample is iteratively sent to its children. Wang et al. [2017] introduced a reject option by considering two specific local risk minimization problems in each node of the hierarchy. By starting at the root node, the tree is recursively traversed until an internal or leaf node is returned as prediction.

In hierarchical classification, many authors have considered the optimization of hierarchical loss functions, which evaluate the hierarchical distance between the predicted node and the ground truth node – see [Bi and Kwok, 2015] for an overview. Those approaches also return a single node of the hierarchy as prediction, so they restrict the representation complexity to 1 as well. An exception worth mentioning is Oh [2017], where the so-called top-$k$ hierarchical loss is introduced, which extends the hierarchical loss function proposed by Cesa-bianchi et al. [2004] to the top-$k$ setting. This method has no constraint on the representation complexity. Similarly, Mortier et al. [2021] also consider a factorization like Eqn. (1) without any constraint on the representation complexity, but here set-based utility functions are optimzed. Yang et al. [2017a] also evaluate different set-based utility functions in a framework where hierarchies are considered for computational reasons.

Finally, due to the reduction in (4), our problem could be reduced to the knapsack problem with conflict graph. There are also some correspondences with the maximum independent set problem and the maximum vertex weight clique problem [Bettinelli et al., 2017, Gurski and Rehs, 2019, Pferschy and Schauer, 2017, Vassilevska, 2009, Wang et al., 2016]. Those problems have been extensively studied in the literature, and depending on the problem statement, different algorithms have been proposed. Generally speaking, the knapsack problem is an NP-hard problem class in combinatorial optimization. However, exact and approximate pseudo-polynomial algorithms, based on dynamic programming and branch-and-bound, exist for special cases of conflict graphs, such as co-graphs or graphs with bounded clique width [Gurski and Rehs, 2019, Pferschy and Schauer, 2017, Bettinelli et al., 2017]. However, in addition to the structure of our conflict graph, it is not immediately clear whether our problem statement allows a dynamic programming solution, since an additional constraint on the representation complexity is considered in problem (3). This additional constraint is atypical for classical KCG problems. Therefore, a more thorough analysis on the structure of the conflict graph in problem (3) and a translation to more efficient algorithms appear to be interesting problems for future work.

Table 1: Overview of of image (top) and text (bottom) datasets used in the experiments. Notation: $K$ – number of classes, $D$ – number of features, $N$ – number of samples

| Dataset | K | D | $N_{train}$ | $N_{test}$ |
|---|---|---|---|---|
| **Caltech-101** [Li et al., 2003] | 97 | 1000 | 4338 | 4339 |
| **Caltech-256** [Griffin et al., 2007] | 256 | 1000 | 14890 | 14890 |
| **PlantCLEF2015** [Goëau et al., 2015] | 1000 | 1000 | 91758 | 21447 |
| **Bacteria** [RIKEN, 2013] | 2659 | 1000 | 10587 | 2294 |
| **Proteins** [Li et al., 2018] | 3485 | 1000 | 11830 | 10179 |

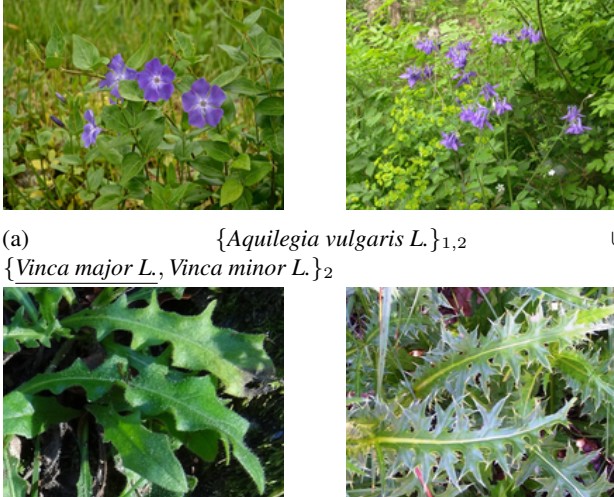

(a) $\qquad$ {*Aquilegia vulgaris L.*}$_{1,2}$ $\qquad$ $\cup$ {*Vinca major L.*, *Vinca minor L.*}$_2$

(b) $\qquad$ {*Carduus defloratus L.*}$_{1,2,3}$ $\qquad$ $\cup$ {*Carduus nigrescens Vill.*}$_{2,3}$ $\cup$ {*Leontodon hispidus L.*}$_3$

Figure 2: Left: image of *Vinca major L.* (top) and *Leontodon hispidus L.* (bottom) from PlantCLEF2015 with corresponding predictions. Set sizes were restricted to five and for each example, different representation complexities were considered. Notation: $\{\ldots\}_{i,j} :=$ set that is predicted when restricting the representation complexity by $i$ and $j$. Right: image of corresponding top-1 prediction, in this case *Aquilegia vulgaris L.* (top) and *Carduus defloratus L.* (bottom).

## 5 EXPERIMENTS

We perform two types of experiments. In a first experiment, we illustrate the usefulness of restricting the representation complexity on a fine-grained visual categorization dataset. In a second experiment, we compare the different algorithms that we propose with some baselines, by looking at predictive performance and runtime efficiency for five different benchmark datasets. Summary statistics related to the datasets can be found in Table 1. For all datasets, we use a predefined hierarchy that was provided with the data. For detailed information, related to the experimental setup, we refer the reader to App. B.

Table 2: Performance versus runtime for MVM, TOP-$k$, KCG, RTS and SVBOP-HF on five benchmark datasets. For all models, we consider different restrictions for the representation complexity $r$ and set size $k$. Notation: $t_{\text{train}}$ – CPU training time in seconds per training instance, Acc. – test accuracy for underlying probabilistic model, $t_{\text{test}}$ – CPU top-1 prediction time in seconds per test instance, R – avg. recall on test set, $|\hat{Y}|$ – avg. prediction size on test set, $t$ – CPU prediction time in seconds per test instance, $n$ – complexity per test instance (see main paper for more information).

| DATASET | MODEL-$r$ | $t_{\text{train}}$ | ACC. | $t_{\text{test}}$ | $R$ | $|\hat{Y}|$ | $t$ ($k=5$) | $n$ | $R$ | $|\hat{Y}|$ | $t$ ($k=10$) | $n$ |
|---|---|---|---|---|---|---|---|---|---|---|---|---|
| CALTECH-101 | MVM-1 | | | | 0.9215 | 2.3713 | 0.0020 | 117 | 0.9303 | 4.4690 | 0.0020 | 122 |
| | MVM-2 | 0.0013 | 0.8993 | 0.0006 | 0.9602 | 3.1725 | 0.0103 | 6359 | 0.9669 | 5.3850 | 0.0114 | 7245 |
| | MVM-3 | | | | 0.9734 | 3.9037 | 0.2892 | 222711 | 0.9780 | 6.5212 | 0.3587 | 278537 |
| | TOP-$k$ | | | | 0.9831 | 5.0000 | 0.0007 | - | 0.9926 | 10.0000 | 0.0008 | - |
| | KCG-1 | | | | 0.9113 | 2.5674 | 0.0053 | | 0.9183 | 5.4093 | 0.0053 | |
| | KCG-2 | 0.0013 | 0.8919 | 0.0006 | 0.9558 | 3.1931 | 0.0056 | $425 \times 128$ | 0.9623 | 5.9845 | 0.0056 | $425 \times 128$ |
| | KCG-3 | | | | 0.9729 | 3.8329 | 0.0053 | | 0.9764 | 7.0317 | 0.0057 | |
| | KCG-$\infty$ | | | | 0.9838 | 4.4481 | 0.0053 | | 0.9931 | 8.9662 | 0.0057 | |
| | RTS-1 | | | | 0.9076 | 2.5550 | 0.0008 | 3.9090 | 0.9100 | 5.4079 | 0.0007 | 3.4421 |
| | RTS-2 | 0.0022 | 0.8898 | 0.0007 | 0.9468 | 3.4234 | 0.0009 | 8.3639 | 0.9579 | 6.1926 | 0.0009 | 9.1150 |
| | RTS-3 | | | | 0.9609 | 4.1470 | 0.0010 | 12.7731 | 0.9706 | 7.5350 | 0.0011 | 15.5488 |
| | SVBOP-HF | | | | 0.9729 | 5.0000 | 0.0010 | - | 0.9885 | 10.0000 | 0.0011 | - |
| CALTECH-256 | MVM-1 | | | | 0.7705 | 1.8499 | 0.0043 | 284 | 0.8016 | 5.1747 | 0.0043 | 303 |
| | MVM-2 | 0.0013 | 0.7581 | 0.0006 | 0.8569 | 3.1443 | 0.0550 | 39602 | 0.8774 | 6.9616 | 0.0627 | 44917 |
| | MVM-3 | | | | 0.8882 | 3.8196 | 4.2796 | 3385995 | 0.9040 | 7.9964 | 5.4916 | 4301775 |
| | TOP-$k$ | | | | 0.9196 | 5.0000 | 0.0007 | - | 0.9515 | 10.0000 | 0.0007 | - |
| | KCG-1 | | | | 0.7747 | 1.8688 | 0.0082 | | 0.8034 | 5.1944 | 0.0083 | |
| | KCG-2 | 0.0012 | 0.7625 | 0.0006 | 0.8611 | 3.1744 | 0.0082 | $1140 \times 318$ | 0.8789 | 6.9935 | 0.0086 | $1140 \times 318$ |
| | KCG-3 | | | | 0.8918 | 3.8339 | 0.0085 | | 0.9077 | 7.9933 | 0.0088 | |
| | KCG-$\infty$ | | | | 0.9214 | 4.9950 | 0.0081 | | 0.9519 | 9.9709 | 0.0087 | |
| | RTS-1 | | | | 0.6955 | 1.8809 | 0.0008 | 4.6238 | 0.7181 | 5.2998 | 0.0008 | 4.0122 |
| | RTS-2 | 0.0023 | 0.6640 | 0.0008 | 0.7832 | 3.1265 | 0.0009 | 7.9226 | 0.8087 | 7.1283 | 0.0009 | 9.0010 |
| | RTS-3 | | | | 0.8192 | 3.8171 | 0.0010 | 11.2637 | 0.8445 | 8.1210 | 0.0011 | 15.1166 |
| | SVBOP-HF | | | | 0.8576 | 5.0000 | 0.0010 | - | 0.9079 | 10.0000 | 0.0012 | - |
| PLANTCLEF2015 | MVM-1 | 0.0013 | 0.4938 | 0.0006 | 0.5220 | 2.0595 | 0.0149 | 1571 | 0.5536 | 3.9500 | 0.0148 | 1613 |
| | TOP-$k$ | | | | 0.7239 | 5.0000 | 0.0007 | - | 0.7969 | 10.0000 | 0.0007 | - |
| | KCG-1 | | | | 0.5236 | 2.1305 | 0.0708 | | 0.5547 | 4.1527 | 0.0707 | |
| | KCG-2 | 0.0012 | 0.4949 | 0.0006 | 0.6226 | 3.2944 | 0.0716 | $4158 \times 1641$ | 0.6538 | 6.1007 | 0.0725 | $4158 \times 1641$ |
| | KCG-3 | | | | 0.6690 | 3.9379 | 0.0746 | | 0.7003 | 7.2684 | 0.0755 | |
| | KCG-$\infty$ | | | | 0.7187 | 4.9743 | 0.0752 | | 0.7923 | 9.9064 | 0.0789 | |
| | RTS-1 | | | | 0.4645 | 2.1577 | 0.0009 | 3.1423 | 0.5004 | 4.2118 | 0.0009 | 2.7745 |
| | RTS-2 | 0.0033 | 0.4278 | 0.0007 | 0.5642 | 3.3311 | 0.0011 | 6.5725 | 0.6001 | 6.1405 | 0.0010 | 6.8671 |
| | RTS-3 | | | | 0.6099 | 4.0115 | 0.0012 | 10.3591 | 0.6432 | 7.3037 | 0.0012 | 12.3894 |
| | SVBOP-HF | | | | 0.6626 | 5.0000 | 0.0011 | - | 0.7433 | 10.0000 | 0.0013 | - |
| BACTERIA | MVM-1 | 0.0001 | 0.5704 | 0.0000 | 0.6215 | 2.0788 | 0.0377 | 3994 | 0.6976 | 4.2610 | 0.0380 | 4096 |
| | TOP-$k$ | | | | 0.8063 | 5.0000 | 0.0001 | - | 0.8675 | 10.0000 | 0.0002 | - |
| | KCG-1 | | | | 0.6369 | 1.9423 | 1.0533 | | 0.7038 | 4.1752 | 1.0542 | |
| | KCG-2 | 0.0001 | 0.5929 | 0.0000 | 0.7205 | 3.4833 | 1.0611 | $29556 \times 4330$ | 0.7812 | 5.7421 | 1.0646 | $29556 \times 4330$ |
| | KCG-3 | | | | 0.7606 | 4.1175 | 1.0606 | | 0.8081 | 7.3407 | 1.0855 | |
| | KCG-$\infty$ | | | | 0.7931 | 5.0000 | 1.0672 | | 0.8741 | 10.0000 | 1.0918 | |
| | RTS-1 | | | | 0.8398 | 1.7742 | 0.0005 | 7.9489 | 0.8913 | 3.7601 | 0.0005 | 7.5678 |
| | RTS-2 | 0.0030 | 0.8006 | 0.0003 | 0.9353 | 3.1959 | 0.0005 | 10.5867 | 0.9516 | 5.5599 | 0.0006 | 11.1743 |
| | RTS-3 | | | | 0.9608 | 3.8191 | 0.0006 | 13.4784 | 0.9705 | 6.6738 | 0.0006 | 15.7210 |
| | SVBOP-HF | | | | 0.9802 | 5.0000 | 0.0006 | - | 0.9952 | 10.0000 | 0.0007 | - |
| PROTEINS | MVM-1 | 0.0000 | 0.7699 | 0.0000 | 0.7766 | 1.3152 | 0.0489 | 3626 | 0.7829 | 2.2505 | 0.0500 | 3672 |
| | TOP-$k$ | | | | 0.9009 | 5.0000 | 0.0001 | - | 0.9235 | 10.0000 | 0.0002 | - |
| | KCG-1 | | | | 0.7728 | 1.3245 | 0.4748 | | 0.7802 | 2.3300 | 0.4739 | |
| | KCG-2 | 0.0000 | 0.7667 | 0.0000 | 0.8439 | 2.3042 | 0.4758 | $14784 \times 3792$ | 0.8494 | 4.2730 | 0.4751 | $14784 \times 3792$ |
| | KCG-3 | | | | 0.8734 | 3.2057 | 0.4837 | | 0.8765 | 5.8075 | 0.4861 | |
| | KCG-$\infty$ | | | | 0.9003 | 4.9320 | 0.4888 | | 0.9219 | 9.8309 | 0.4906 | |
| | RTS-1 | | | | 0.7936 | 1.3045 | 0.0004 | 5.0570 | 0.8012 | 2.2052 | 0.0003 | 4.8834 |
| | RTS-2 | 0.0016 | 0.7806 | 0.0002 | 0.8610 | 2.3161 | 0.0004 | 7.2716 | 0.8664 | 3.6366 | 0.0005 | 7.7215 |
| | RTS-3 | | | | 0.8842 | 3.2457 | 0.0005 | 9.0939 | 0.8885 | 4.7484 | 0.0006 | 10.9509 |
| | SVBOP-HF | | | | 0.9086 | 5.0000 | 0.0005 | - | 0.9308 | 10.0000 | 0.0007 | - |

## 5.1 SOME ILLUSTRATIONS

We illustrate the usefulness of our framework on the Plant-CLEF2015 dataset. This is a well-known image dataset with fine-grained annotations of 1000 plant species. The dataset is characterized by a substantial class ambiguity, making accurate predictions on the species level often impossible. In Fig. 2, we show two images (left) and the predictions for the labels. Additionally, we also show images of corresponding top-1 predictions (right). The subscript $i$ means that the subset belongs to the prediction obtained by restricting the representation complexity by $i$. For the top image, an example of the *Vinca major L.* class, we show two predictions obtained by restricting the representation complexity to one and two, respectively. If the representation complexity is two, then the ground truth class is included in the solution. Class ambiguity is present at a higher level in the plant species hierarchy, since both the genera *Aquilegia* and *Vinca* contain plants with similar flowers, which are difficult to distinguish from each other, as can be observed by comparing the left with the right image. In this case, predicting a single node from the hierarchy (i.e., by restricting the representation complexity to one) would not be sufficient, given the restriction on the set size. For the bottom image, an example of class *Leontodon hispidus L.*, we even have a higher degree of ambiguity, which is illustrated by the fact that we need a representation complexity of three for the ground truth to be included in the predicted set.

## 5.2 BENCHMARKING RESULTS

In a second set of experiments, with results shown in Table 2, we analyse the performance versus runtime for MVM, KCG and RTS on the five benchmark datasets. In addition, we also include results for two baselines from literature: (i) the pointwise size control framework, as described by Chzhen et al. [2021], which corresponds to top-$k$ prediction by using a flat probabilistic model (TOP-$k$), and (ii) SVBOP-HF, an exact inference algorithm that was proposed by Mortier et al. [2021] for top-$k$ prediction by using a probabilistic model with hierarchical factorization. Note that the latter baselines are only applicable when we don't have a restriction on the representation complexity (i.e., $r = \infty$ in Table 2). More precisely, for SVBOP-HF and RTS, we use a hierarchical softmax layer, as given by Eqn. (1), whereas for MVM, TOP-$k$ and KCG, we use a (flat) softmax layer for the probabilistic model. In a first step, we train and validate our probabilistic model on the training set. Finally, in a last inference step, we use our trained probabilistic model to obtain predictions on the test set. For KCG, we tested different mixed-integer solvers such as SCIP, CBC and a long-step dual simplex solver from the GLPK kit [Achterberg, 2009, Forrest et al., 2018, Makhorin, 2001]. However, we only mention the results for the GLPK solver, since for this solver the runtime was substantially lower for all experiments.

For each experiment, we show the training time in seconds per instance $t_{\text{train}}$, the accuracy of the underlying probabilistic model Acc., time in seconds to obtain the top-1 prediction for a test instance $t_{\text{test}}$, average recall on test set $R$, average prediction size on test set $|\hat{Y}|$ and prediction time in seconds per test instance $t$. In addition, we also analyse the complexity for each test instance by means of a method-specific complexity metric $n$, which corresponds to the size of the feasible set $\mathcal{M}_{r,k}$, dimensionality of the matrix $A_{\mathcal{T}}$ and the number of nodes that are popped from the priority queue in line 3 of Alg. 4. In terms of runtime efficiency, RTS significantly outperforms MVM and KCG for all datasets. This is also illustrated by looking at the complexity metrics. For the biological datasets, we only considered a representation complexity of 1 for MVM, since higher values for $r$ quickly gave rise to out-of-memory usage errors due to the size of the matrix $M$ increasing exponentially. In general, the improvement in runtime for RTS comes with a cost of lower performance of the underlying probabilistic model. Only for the non-visual biological datasets, there seems to be an improvement when a hierarchy is considered. Perhaps, this finding can be explained by the fact that taxonomic information is much more present in those datasets, compared to the image datasets. Finally, increasing the representation complexity generally results in a higher recall and set size, which once again illustrates its usefulness. In extremis, when the representation complexity is not restricted, the best performance is observed. However, in that case, the complexity of our prediction is also much higher, which is not really meaningful in case we want to restrict predictions to a predefined hierarchy.

## 6 CONCLUSION

In this work, we proposed a new decision-theoretic framework for set-valued prediction in hierarchical classification by introducing the notion of representation complexity. This complexity allows the user to relax the often strong restriction that is implied by hierarchical classification, namely that predictions should correspond to single nodes of a predefined hierarchy. We proposed several algorithms that solve the challenging optimization problem in an exact way. One of those algorithms, based on a recursive tree search method that uses a hierarchical factorization of the conditional class distribution, shows especially promising results in terms of runtime complexity. An interesting future direction could be to generalize our framework to other settings that are commonly found in the set-valued prediction literature, such as pointwise and average control of the set size or error rate. Moreover, the translation of our problem to the well-known knapsack problem with conflict graph seems interesting and opens the potential to improve the runtime complexity of the recursive tree search method by exploiting the specific structure of our conflict graph.

## Acknowledgements

For this work W.W. received funding from the Flemish government under the "Onderzoeksprogramma Artificiële Intelligentie (AI) Vlaanderen" Programme (Number 174L00121).

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
