# OpenReview forum: "Set-valued prediction in hierarchical classification with constrained representation complexity"
_auai.org/UAI/2022/Conference — UAI 2022 Poster_

### Official Review · Reviewer_CHRs · 2022-03-26

**Q2(1) Originality/Novelty:** 3
**Q2(2) Significance/Impact:** 2
**Q2(3) Correctness/Technical Quality:** 4
**Q2(6) Clarity Of Writing:** 4
**Q6 Overall Score:** 5
**Q8 Confidence In Your Score:** 4

**Q1 Summary And Contributions:**

The paper proposed a way to generate a  probability over a subset of labels. The work assumes that labels have hierarchical structures, and the subset of labels to be predicted should 1) have large likelihood to emphasize on recalls, 2) contains limited labels to be informative, 3) have low representation complexity to be concise. Compared to existing work, authors designed a metrics on describing the representation complexity and proposed multiple exact solver to find the best prediction.

**Q2 Assessment Of The Paper:**

More detailed information regarding each of these aspects is given below:

**Q2(4) Quality Of Experiments (Optional):**

3: Good: The experimental evaluation is adequate, and the results convincingly support the main claims.

**Q2(5) Reproducibility:**

4: Excellent: Key resources (e.g., proofs, code, data) are available and key details (e.g., proof sketches, experimental setup) are comprehensively described for competent researchers to confidently and easily reproduce the main results.

**Q3 Main Strengths:**

 The paper proposed a simple idea to model the prediction complexity.  The idea is intuitive and well motivated. In addition, author proposed some non-trivial exact solutions on finding the best prediction under the constraint of the prediction complexity and prediction cardinality. The exact solution RTS shows significant inference speed up compared to the other solution that relies on existing ILP solvers.


**Q4 Main Weakness:**

The empirical evaluation does not show the advantage of the new method compared to existing ones. From table 2, it shows that the recall of the new method is worse than the baselines SVBOP-HF and Top-k. It would be nice if author could demonstrate what we gain by suffering the loss on the recall. For example, what is the average expression complexity of the solutions that are given by the SVBOP-HF and Top-k?

**Q5 Detailed Comments To The Authors:**

1) Is Equation 1 a constraint on the multi-label distribution? Cannot we obtain this factorization from any flat distribution and vice versa?

2) In the evaluation, can author explain why KCG-inf not always select k element as the prediction when there is no expression complexity penalty?

3) In the evaluation, can author also explain why MVM-1 and KCG-1 RTS-1 not having the same quality result? Aren't they all solving the same optimization problem exactly? Shouldn't they have similar quality metrics, i.e. Recall ?

**Q7 Justification For Your Score:**

The paper is very clearly written and the idea is very simple and sound. My major concern is the significance of the work. The evaluation does not show enough evidence that the expression complexity improves the quality of the predicted subset in practice.

**Q9 Complying With Reviewing Instructions:**

1: Yes.

---

### Official Review · Reviewer_R2x1 · 2022-04-11

**Q2(1) Originality/Novelty:** 3
**Q2(2) Significance/Impact:** 2
**Q2(3) Correctness/Technical Quality:** 4
**Q2(6) Clarity Of Writing:** 4
**Q6 Overall Score:** 6
**Q8 Confidence In Your Score:** 4

**Q1 Summary And Contributions:**

The authors propose a new method to produce complexity-constrained predictions in hierarchical classification.
The authors posit a new definition of set prediction representation complexity.
The authors provide three different algorithms to actually select a prediction set with maximum posterior within a complexity constraint budget.

The authors demonstrate that their method is competitive while their third proposed algorithm is fastest.

**Q2 Assessment Of The Paper:**

More detailed information regarding each of these aspects is given below:

**Q2(4) Quality Of Experiments (Optional):**

3: Good: The experimental evaluation is adequate, and the results convincingly support the main claims.

**Q2(5) Reproducibility:**

3: Good: Key resources (e.g., proofs, code, data) are available and key details (e.g., proofs, experimental setup) are sufficiently well-described for competent researchers to confidently reproduce the main results.

**Q3 Main Strengths:**

The paper is very well written and manages to expose interesting and new approaches to the set prediction problem.

The three algorithms the authors present are well exposed, well principled and the proofs of correctness are convincing.

The practical experiments are also well conducted and varied enough however, the authors are clear about how they show both strengths and limitations to representation complexity constrained set predictions.

**Q4 Main Weakness:**

The main weakness I see with the paper is in its scope of application. I am not sure how many applications will practically benefit from limiting representation complexity which comes at the cost of performance.

**Q5 Detailed Comments To The Authors:**

Abstract:
Maybe mention that set classification also occurs in recommender system kind of approaches (https://arxiv.org/pdf/1812.02353.pdf).

Introduction:
"A lot of classes": please give order of magnitude.
Put emphasis on the limitation that the fact that the classification hierarchy has to known beforehand. For domains with dynamical topics such as news understanding or recommendations, this is usually a major issue.

Page 2 column 1: "On the contrary ... might be more useful and require less effort". What do the authors mean by effort?

Could the authors make it clearer, throughout the paper, whether the improvements of performance they mention are in speed or accuracy?

In Table 2 page 7, could the authors separate out their contributions from others' baselines more clearly?

**Q7 Justification For Your Score:**

I think the paper is a good one but I am a bit worried as to its practical impact scope.

**Q9 Complying With Reviewing Instructions:**

1: Yes.

---

### Official Review · Reviewer_jGj9 · 2022-04-12

**Q2(1) Originality/Novelty:** 3
**Q2(2) Significance/Impact:** 3
**Q2(3) Correctness/Technical Quality:** 3
**Q2(6) Clarity Of Writing:** 2
**Q6 Overall Score:** 5
**Q8 Confidence In Your Score:** 2

**Q1 Summary And Contributions:**

The paper focuses on hierarchical prediction problems and formalises a new inference framework that combines individual class predictions into consistent most likely predictions. The paper introduces a formalisation of a problem as a constraint optimisation problem and several algorithms solving the problem.

**Q2 Assessment Of The Paper:**

More detailed information regarding each of these aspects is given below:

**Q2(4) Quality Of Experiments (Optional):**

3: Good: The experimental evaluation is adequate, and the results convincingly support the main claims.

**Q2(5) Reproducibility:**

3: Good: Key resources (e.g., proofs, code, data) are available and key details (e.g., proofs, experimental setup) are sufficiently well-described for competent researchers to confidently reproduce the main results.

**Q3 Main Strengths:**

Interesting formulation to a known problem
Clear benefit when single-class classifiers are uncertain
Well performing method

**Q4 Main Weakness:**

Writing can be significantly improved
Difficult to see the trend in results

**Q5 Detailed Comments To The Authors:**

This paper is way out of my expertise, so I will focus on high-level comments.

I think the idea the paper proposes is very interesting and it makes sense. It presents a novel look at the problem from the perspective of constraint satisfaction, making a trade-off between the complexity of the set containing predictions and their accuracy.
In my understanding, this is a novel idea that hasn't been explored. However, I am not familiar with the state of the art.

The methods seem to be performing quite well. However, I find it difficult to fully appreciate the algorithms and how they actually solve the problem (e.g., the MILP formulation). A more intuitive explanation of the algorithms would be much appreciated.

Several aspects of the framework didn't sit with me well:

- What is the expectation of the probabilities between different levels of the hierarchy? Considering the example in Figure 1 and the correct label being 3. What is the expected relationship between P(v6) and P(v3)? If 3 is the correct label, should P(v6) and P(v3) be equal, or should the label at the level closest to the correct label be the highest probability? Leaving these expectations out of the discussion in a probabilistic modelling paper leaves too many things ambiguous.

- The paper emphasises the need to be able to solve problems for big r. However, is that really a reasonable expectation? In the end, we want confident models, which should have a low r if I understand correctly?

Please think of a better way to show the results. Such a massive table with so many numbers is unreadable. It is impossible to notice the trend in overall results.

Several parts of the paper could have been written with more care.  For instance, the equation for S_T took me a long time to understand because of the formating. Similarly, R_T^2 and M_{k,r} took a lot of time to understand for something so simple. Also, Algs 2 and 3 are pointless.



**Q7 Justification For Your Score:**

To the best of my understanding, this is a solid paper with a few minor weaknesses. However, my knowledge of the field is very limited.

**Q9 Complying With Reviewing Instructions:**

1: Yes.

---

### Decision · Program_Chairs · 2022-05-15

**Decision:**

Accept (Poster)

**Comment:**

Meta Review: Pros:
1. The paper makes non-trivial advances over the current state-of-the-art.
2. Key resources (e.g., proofs, code, data) are available and key details (e.g., proofs, experimental setup) are sufficiently well-described for competent researchers to confidently reproduce the main results.
3. The experimental evaluation is adequate, and the results convincingly support the main claims.
4. The paper is likely to have moderate to high impact within a subfield of AI.
5. The paper is very well written and manages to expose interesting and new approaches to the set prediction problem.
Cons:
1. Limited empirical evaluations with flat classifiers.
2. Scope of application may be limited. The practical benefits from limiting representation complexity which comes at the cost of performance may be limited.
3. The empirical evaluation does not show the advantage of the new method compared to existing ones.
The authors in their rebuttal do address these cons (convincingly to this Area Chair).